# rTMS Reduces Craving and Alcohol Use in Patients with Alcohol Use Disorder: Results of a Randomized, Sham-Controlled Clinical Trial

**DOI:** 10.3390/jcm11040951

**Published:** 2022-02-11

**Authors:** Maarten Belgers, Philip Van Eijndhoven, Wiebren Markus, Aart H. Schene, Arnt Schellekens

**Affiliations:** 1IrisZorg, Center for Addiction Care and Sheltered Housing, Mr. B.M. Teldersstraat 7, 6842 CT Arnhem, The Netherlands; w.markus@iriszorg.nl; 2Nijmegen Institute of Scientist-Practitioners in Addiction (NISPA), Radboud University Nijmegen, 6525 GA Nijmegen, The Netherlands; 3Department of Psychiatry, Radboud University Medical Center, Geert Grooteplein Zuid 10, 6525 GA Nijmegen, The Netherlands; philip.vaneijndhoven@radboudumc.nl (P.V.E.); arnt.schellekens@radboudumc.nl (A.S.)

**Keywords:** transcranial magnetic stimulation, alcohol use disorder, relapse, abstinence, craving, neuromodulation

## Abstract

(1) Background: Current evidence-based treatments for alcohol use disorder (AUD) are moderately effective. Studies testing repetitive transcranial magnetic stimulation (rTMS) in AUD commonly apply a limited number of rTMS sessions with different rTMS settings, showing inconsistent effects on craving for alcohol. This study tested the efficacy of a robust rTMS protocol on craving and alcohol use. (2) Methods: In a single-blind randomized controlled trial in recently detoxified patients with AUD, ten days of high-frequency rTMS over the right dorsolateral prefrontal cortex on top of treatment as usual (*n* = 14) was compared with sham rTMS (*n* = 16). Outcome measures were alcohol craving and use over a follow-up period of one year. Analysis was performed by means of repeated measures multivariate analysis of variance. (3) Results: The results showed a main group-by-time interaction effect on craving (Wilks’ Λ = 0.348, F (12, 17) = 2.654, *p* = 0.032) and an effect of group on alcohol use (Wilk’s Λ = 0.44, F (6, 23) = 4.9, *p* = 0.002), with lower alcohol craving and use in the group with active rTMS compared to the control group. Differences in craving between groups were most prominent three months after treatment. At 12 months follow-up, there was no effect of rTMS on craving or abstinence. (4) Conclusions: This small-scale randomized controlled trial showed the efficacy of high-frequency rTMS over the right dlPFC diminished alcohol craving and use in recently detoxified patients with AUD during the first months after detoxification. These findings suggest that rTMS might be an effective add-on in treating patients with AUD and warrant replication in future large-scale studies.

## 1. Introduction

Alcohol use disorder (AUD) is a chronic relapsing disorder characterized by an impaired ability to control alcohol use, leading to clinically significant impairment or distress [1]. A core symptom of AUD is craving, which is associated with relapse after treatment [2]. The prevalence of AUD in Europe is as high as 14.6% in adult men and 3.5% for women [3]. AUD is associated with a nearly 6-fold increase in all-cause mortality [4]. Loss of disability-adjusted life years for AUD in Europe ranks highest for all mental and neurological disorders [5]. Given this tremendous burden of disease, it is important to have effective treatment options for patients with AUD.

Psychosocial (like motivational interviewing and cognitive behavioral therapy) and pharmacological treatments (like disulfiram, acamprosate, and naltrexone) show low to moderate effect sizes [6,7]. With these treatments, relapse rates for patients with AUD are still as high as 60% within one year after reaching abstinence [8,9]. To improve these treatment results, new treatment modalities are urgently needed.

Noninvasive brain stimulation may offer a promising new treatment strategy targeting AUD via a different mechanism than existing treatments [10]. Repetitive transcranial magnetic stimulation (rTMS) is a neuromodulation technique that applies alternating magnetic fields produced by an electromagnetic coil placed over the patient’s skull. These fields induce small, alternating currents in the cortex of the brain. High-frequency rTMS stimulates the underlying brain region with an effect that lasts beyond the duration of the treatment session. Previous studies in substance use disorders have shown that the dorsolateral prefrontal cortex (dlPFC) might be a relevant rTMS target since it plays an important role in behavioral control and shows low activity in patients with AUD [11].

Meta-analyses of neuromodulation studies in patients with various addictive disorders showed a moderate to a large effect size of rTMS in decreasing craving for drugs of abuse [12,13,14,15]. However, more recent meta-analyses were inconclusive because of the heterogeneity of effects of rTMS in the included studies. This heterogeneity might result from variation in the targeted alcohol or drug use disorder, duration, the intensity of rTMS treatment, localization of rTMS brain target, and variation in follow-up duration [10,16]. Therefore, recent studies suggest investigating more robust rTMS protocols with a sufficient dose of rTMS and preferably a minimum of three months follow-up periods [17,18].

Two recently published sham-controlled rTMS studies testing robust, high-frequency rTMS in patients with AUD, however, did not find any effects of rTMS compared to placebo. One study applied deep rTMS on the insula [19] but failed to find supportive evidence for targeting this region in AUD. The other study applied high frequent rTMS over the right dlPFC but only reported effects on impulsivity as an outcome variable [20], making it difficult to establish the clinical relevance of the used rTMS protocol for AUD. 

The current study aimed to test the efficacy of a robust high-frequency rTMS protocol, stimulating the right dlPFC, with clinical outcome measures alcohol craving and use, and a follow-up period of one year. Specifically, we tested the hypotheses that compared to sham rTMS, rTMS over the right dlPFC, added to treatment as usual (TAU), would lead to reduced (1) alcohol craving (primary outcome) and (2) alcohol use (secondary outcome).

## 2. Materials and Methods

### 2.1. Study Design

In a single blind randomized controlled trial efficacy of rTMS was investigated. The research protocol was approved by the Medical Ethical Committee of the Radboud University Medical Centre (protocol nr. NL46974.091.13, Nijmegen, The Netherlands) and registered in a trial Register (ClinicalTrials.gov Identifier: NCT01973127, Bethseda, MD, USA).

### 2.2. Study Sample

Eligible patients with AUD were recruited between 2015 and 2019 at two addiction care centers (IrisZorg and Novadic-Kentron) and Radboud University Medical Centre in The Netherlands. In total, 37 individuals were screened for eligibility, wherefrom 34 were included and randomized (three could no longer be contacted after initial contact). After baseline measurements, four individuals withdrew their consent to participate.

Inclusion criteria were: (1) meeting DSM-5 criteria for AUD as their primary diagnosis using the structured clinical interview for DSM-5 disorders—clinician version (SCID-5-CV) [21] (use of other substances was no reason for exclusion); (2) age between 20 and 65 years; (3) successful recent (<6 weeks) inpatient detoxification of alcohol and (4) written informed consent. Exclusion criteria were: (1) any psychiatric condition that, due to the severity of symptoms, interfered with TAU; (2) standard rTMS contraindications (history of epilepsy, ferromagnetic implants in the head, a history of neurosurgical operations, or a pacemaker implant); (3) use of medication known to substantially lower the threshold of epileptic seizures (e.g., clozapine, pethidine, aminophylline), and (4) other factors which made study-procedures not feasible (most notably intellectual disabilities, major somatic disabilities, insufficient Dutch language proficiency).

### 2.3. Treatment

#### 2.3.1. TAU

All participants received TAU at one of the participating addiction care centers, consisting of outpatient CBT and/or anti-craving medication. Participants received TAU during the total period of the study, including the follow-up period.

#### 2.3.2. rTMS

For applying rTMS, a 70 mm double air film, the figure of eight coils, and a Magstim Rapid2 stimulator were used [22]. First, the target of the rTMS coil was defined as point F4 on the right dlPFC, according to the international 10–20 system for electroencephalography [23]. After defining F4, this point was marked on a cap placed on the head of the participant relative to anatomical landmarks of the patient’s skull (nasion-inion) to reliably target F4 during each following treatment session. Next, the resting motor threshold (MT, the threshold at which motor neurons are stimulated to provoke muscular contraction) was determined by applying single pulses of TMS in steadily increasing intensity over the right motor cortex. When 5 out of 10 stimuli resulted in a muscular contraction in the left lower arm or hand muscles, this stimulation intensity was taken as MT. The actual rTMS treatment was given at an intensity of 110% of the MT. During each rTMS treatment session, participants received sixty 10 Hz trains of 5 s at 110% MT, resulting in 3000 pulses per session (30 min total treatment time) and 30,000 pulses during the total study. This procedure has been proven effective in treating depressive disorders and is used in more recent studies on rTMS in addiction [24]. A total of 30,000 pulses are amongst the highest number used in studies in this field while being well below the threshold of increasing risks on side effects [25,26].

#### 2.3.3. Sham rTMS

The placebo effect of rTMS is potentially large [27]. To address this issue, a sham intervention was incorporated, which is the same as that for rTMS, except that during the sessions, the coil with two wings was rotated 90 degrees relative to the plane of where the coil was placed to the skull in a real rTMS session [28]. Because of the directional properties of the magnetic field, most of the field lines would not enter the skull, and hence, no effect on the cortex would be applied. In this way, the setup procedure, the buzzing of the machine and clicking of the coil, etc.., was noticeable for the participant, largely mimicking a real rTMS treatment. The investigator (first author) was not blinded for the treatment modality. Both the real and the sham groups were given hearing sponges during treatment.

### 2.4. Measurements

#### 2.4.1. Sample and Treatment Characteristics

The following variables were assessed at baseline: age, gender, handedness, IQ (by means of the Dutch version of the Adult Reading Test (NLV) [29], years of education, use of anti-craving, antidepressant, and antipsychotic medication at baseline (yes/no), duration of AUD (years), number of previously followed AUD treatments and presence of psychiatric disorders and other substance use disorders (using the Mini-International Neuropsychiatric Interview (MINI) [30].

#### 2.4.2. Outcome Measurements

Alcohol craving (primary outcome) was measured at five timepoints: at baseline (day one of the rTMS treatment), at the end of the rTMS treatment (10th day), and at follow-up: one, three, and 12 months after finishing rTMS treatment. Three instruments were used at all timepoints to measure alcohol craving:Visual Analog Scale (VAS). The VAS is commonly used in studies to assess the severity of craving in patients with substance use disorder [31]. A total VAS score was defined by the mean of two VAS scores (on a 100 mm line, with anchor points of 0 (not at all/don’t agree at all) and 10 (desperately/totally agree)) on the question “How much do you want a drink at this moment?” and the statement “If I could drink, I would probably do it”.Alcohol Urge Questionnaire (AUQ). The AUQ is a validated instrument (Cronbach α = 0.918; test-retest reliability r = 0,82). It measures momentary alcohol craving in patients with AUD [32,33]. It contains eight items referring to statements such as the desire to drink, the expectation of the desired outcome from drinking, and the inability to avoid drinking if alcohol was available at that moment. Participants indicated their level of agreement on a seven-point scale (range 1–7, with anchor points: “Strongly disagree” and “Strongly agree”). A total score is calculated by summation of the item scores, with reversed scoring for two items. A higher score reflects a higher level of craving.Obsessive-Compulsive Drinking Scale—short version (OCDS-5). The OCDS-5 is a shortened version of the original OCDS [34] (Cronbach α = 0.814). The OCDS-5 is widely used in addiction treatment to measure mean craving over the past seven days [35]. It contains five items in a five-point (0–4) Likert-type scale format [36]. A total score is calculated by summation of the points attributed per item. Higher scores are indicative of higher craving levels.

Indices of alcohol use were measured using different instruments at different timepoints: At baseline (day 1 of the rTMS treatment) and at the start of each rTMS treatment session (2nd–10th day), participants were asked about their alcohol use since the last treatment.At follow-up 1, 3, and 12 months after rTMS treatment, alcohol use was assessed using the TimeLine Follow-Back (TLFB) method over the previous period of 1 month. The TLFB is a validated instrument to systematically estimate alcohol use over a specified timeframe (Spearman’s ρ = 0.93) [37,38]. Participants indicated the number of days they had drunk alcohol and the quantity and type of beverage they had consumed, noted as the quantity of a standard drink (containing 10 mg alcohol).

With these data, sampled over four periods of time, which spanned a total of 12 months, six alcohol use indices were calculated: (1) percentage abstinence at endpoint; (2) total amount of alcohol consumption during measured time periods; (3) mean alcohol consumption per day during measured time periods; (4) time to relapse in days, as defined by relapse in a heavy drinking day (HDD) (defined as more than 60 g alcohol per day for men or more than 40 g of alcohol per day for women [39]. Time to relapse is counted from day one of the treatment up until encountering a relapse during the timeframes of sampling. When participants did not relapse during the study period, the time to relapse was defined as 364 days. Finally, (5) the total number of abstinent days during the study in the sampled timeframes and (6) the total number of HDD during the study in the sampled timeframes were assessed.

### 2.5. Procedure

Medical doctors in addiction care centers were informed about the study. They introduced the study to patients with AUD, who were on the verge of, or recently started with an inpatient alcohol detoxification. When interested, potential participants were informed about the study by the investigator and provided written consent when willing to participate. Next, participants were screened for in/exclusion criteria. After enrolment in the study, participants were randomized to either active rTMS or the sham condition, based on a predetermined randomization sequence with an allocation rate of 1:1. The randomization sequence was computer-generated (randomizer.org) before the start of the data collection.

After allocation to one of the two groups, (baseline) sample characteristics, stimulus location, and rTMS intensity were determined. Next, participants received ten active or sham rTMS sessions over ten consecutive days. Though some patients interrupted rTMS treatment over the weekend, leading to a variation of 10–14 days of the total treatment period. After one, three, and 12 months following the last rTMS session, participants were visited at home or visited a treatment facility to assess relapse rates and craving. For a schematic overview of this procedure, see Figure 1.

### 2.6. Analyses

The Statistical Package for Social Sciences (SPSS) version 27 was used to analyze the data [40]. The method of last observation carried forward (LOCF) was used in case of missing data. *p*-values < 0.05 were considered significant.

#### 2.6.1. Sample and Treatment Characteristics and Outcome Variables at Baseline

Baseline sample characteristics were summarized using descriptive statistics and compared between groups. For categorical variables, comparisons were performed with a Chi-square test, while Fisher’s exact tests were used in case the expected counts were less than 5. A two-sample *t*-test was used for continuous variables in case normality was met (Kolmogorov Smirnov test); otherwise, the non-parametric Mann–Whitney-U test was applied. 

#### 2.6.2. Craving

To analyze the effect of rTMS on craving, after calculating correlations (Kendall’s tau-b), a multivariate one-way repeated measures MANOVA was used, with VAS, OCDS-5, and AUQ total scores as continuous dependent variables, time as a within-subject factor (4 levels), and rTMS treatment as a between-subject factor (2 levels). In case of significant results, post-hoc contrast analyses were performed using linear discriminant analysis and ANOVA’s to identify which craving outcome measures at which specific timepoints contributed to the significant findings. In case of unequal distribution of baseline characteristics, sensitivity analyses were performed.

#### 2.6.3. Alcohol Use

To analyze the effect of rTMS on alcohol use, after calculating correlations (Kendall’s tau-b), a multivariate one-way measure MANOVA was used, with our predefined six indices (percentage abstinence at endpoint, alcohol use (total and mean per day), time to relapse, total amount abstinent days, and total amount of HDD-days) as dependent variables, and rTMS treatment (real versus sham) as between-subject factor. In case of significant results, post-hoc contrast analyses were performed using linear discriminant analysis and T-tests to identify which outcome measures contributed to the significant findings. 

## 3. Results

Thirty participants started the treatment, and all had six or more DSM-5 AUD criteria (severe AUD). All participants’ follow-up data were available (Figure 2), except for one follow-up measurement at 12 months. This patient had died from a study-unrelated disease (lung cancer) after 5 months follow-up, where we used LOCF to fill in this one randomly missing measurement. 

No significant differences in baseline characteristics were found between groups, except for PTSD (Χ^2^(1) = 9.299, *p* = 0.002), with more comorbid PTSD in the sham group (Table 1).

### 3.1. Craving

The one-way repeated measures MANOVA on craving showed a main effect of time (Wilks’ Λ = 0.203, F (12, 17) = 5.575, *p* = 0.001), group (Wilks’ Λ = 0.585, F (3, 26) = 6.156, *p* = 0.003), and an group-by-time interaction effect (Wilks’ Λ = 0.348, F (12, 17) = 2.654, *p* = 0.032), indicating increased craving over time for all participants but less increased craving over time in the rTMS group versus sham. Kendall tau-b was below 0.7. Univariate ANOVA showed interaction effects of time x group for all outcome measures, except the OCDS-5 (see Figure 3 and Appendix A). Testing differences between the two groups at each time point showed effects at 1 and 3 months as being most prominent at 3 months (see Appendix A). Sensitivity analysis excluding patients with PTSD at baseline (due to unequal distribution) showed similar findings (see Appendix A). Although the missing data from one patient at one time in our study could be attributed to a random event, we also analyzed data with a worst-case scenario (imputing the highest craving score measured in all individuals during the whole study) not affect the overall results.

### 3.2. Alcohol Outcome Measurements

Because the correlation between alcohol use per day and the total amount of alcohol used was very high (Kendall’s tau-b = 0.994, *p* < 0.01), we combined these measures in analysis. The one-way MANOVA on alcohol use showed an effect of group (Λ = 0.46, F (5, 24) = 5.6, *p* = 0.001), indicating decreased alcohol use in the rTMS group, compared to sham. Post-hoc analysis showed that the rTMS group consumed less alcohol (factor 1.86), had less DD (average difference: 40 DD), and had less HDD (average difference: 21 HDD). The percentage abstinence at the endpoint did not differ between groups (see Appendix A). 

### 3.3. Side Effects

No serious side effects of the treatment were reported or observed. Some participants experienced the treatment as uncomfortable due to muscle twitches around the eye.

## 4. Discussion

This study investigated the effect of rTMS on craving and alcohol use in recently detoxified patients with AUD in a single blind randomized controlled trial. Over a one-year follow-up period, rTMS reduced craving and alcohol use compared to sham rTMS. Differences in craving between groups were most prominent three months after treatment. These findings suggest that rTMS is a safe treatment that might be of added value in treating AUD patients by reducing craving and alcohol use. 

The current findings on rTMS in AUD patients align with previous studies showing the clinical effectiveness of rTMS on alcohol craving and use (for review see: [41]). However, several studies did not show an effect of rTMS, potentially due to shorter treatment duration [42] and follow-up period [43] and different targeted brain regions [44,45]. The current study underlines the importance of a robust stimulation protocol with a sufficiently large number of pulses being applied, with a sufficiently long follow-up duration, and the right dlPFC as a favorable target region. The fact that effects of rTMS were observed both on craving and various indices of alcohol use suggests robustness of the effect, despite limited sample size. Future studies should confirm these exploratory findings in substantially large studies, using drinking levels and craving as by the European Medicines Agency (EMA, Amsterdam, The Netherlands) and the U.S. Food and Drug Administration (FDA, Silver Spring, MA, USA) approved primary and secondary outcome measures, respectively.

The current findings with a higher dose of pulses applied throughout the treatment course suggest persisting effects of rTMS on alcohol craving and use for at least three months. This is in line with the limited studies about rTMS in patients with AUD [46] and with studies showing persisting beneficial effects of rTMS in patients with depressive disorders [47,48].

The decrease of statistical group differences at one year might suggest fading effectiveness of rTMS over time. Indeed, in the depression literature, persisting effects beyond three months have mainly been observed in studies using even more intensive treatment procedures (daily sessions for 6 weeks) or applying booster sessions after an initial treatment episode [49]. However, our sample might have been too small to detect group differences beyond three months of follow-up due to increasing variance in outcome measures. Future studies should assess whether prolonged treatment protocols of more than ten sessions or booster sessions after an initial rTMS treatment episode might increase the long-term efficacy of rTMS in patients with AUD. 

In the current study, the only craving measure that did not show an effect of rTMS was the OCDS-5. The lack of findings on the OCDS-5 might be explained by its seemingly lesser validity when used in a population of heavy drinking subjects, such as in this study [50]. Since the other two craving measures did show clear effects of rTMS on craving, this might indicate that the OCDS is not sensitive enough to detect treatment effects in clinical trials in patients with severe AUD, as found in previous literature [51]. 

Similarly, the observed absolute difference in abstinence at the endpoint failed to reach significance, potentially due to too small a sample size to detect differences in a dichotomous outcome measure. Yet, again this might also indicate that the effects of 10 rTMS sessions did not persist for a year, as outlined above. It might also be that rTMS does reduce the level of alcohol consumption but does not facilitate full abstinence. Indeed, several studies have shown differential effects of evidence-based AUD treatments on alcohol consumption, HDD, and full abstinence [52,53,54]. There is increasing awareness that reduced drinking is a viable option for at least some patients with AUD, especially when full abstinence might not be achievable [55]. Future studies with larger samples and more intense rTMS protocols (>10 sessions) should explore the potential of rTMS for reaching prolonged abstinence versus reduced alcohol consumption. 

In addition, other forms of TMS (like Theta Burst Stimulation) and other neuromodulation techniques (like transcranial Direct Current Stimulation) should be more extensively evaluated in the field of addiction since studies have revealed potential benefit over conventional rTMS or similar effect sizes, respectively [15,56]. Finally, future studies might also consider the potential effect of modifying the role of some personality traits on the effects of rTMS, as observed in depression [57,58,59]. To our knowledge, this has not yet been addressed in patients with AUD. 

In the current study, rTMS was targeted at the right DLPFC. This is in contrast with the majority of depression literature, where the left DLPFC is considered the primary brain target for high-frequency rTMS. However, AUD studies targeting rTMS to the left DLPFC have largely yielded negative findings [60]. Studies in healthy controls have shown that stimulation by rTMS of the right dlPFC strengthens top-down control of aversive stimuli [61,62]. This is in accordance with the hypothesis that the right hemisphere is dominant for the processing of (particularly negative) emotional stimuli [63] and inhibitory control [64]. Furthermore, the neurotoxic effects of alcohol seem more pronounced in the right hemisphere [65]. It could thus be hypothesized that rTMS stimulation on the right dlPFC has restorative effects on cognitive control in AUD patients. 

One study showed that stimulating rTMS on the right dlPFC strengthened connectivity between frontoparietal regions and the striatum in healthy individuals, whereas rTMS on the left dlPFC weakened these connections [66]. The striatum is a core region in reward processing, and a vast body of literature shows striatal abnormalities in patients with substance use disorders, including AUD [67]. Furthermore, the striatum has been attributed a major role in cue reactivity and craving [68]. Strengthening cortico-striatal connectivity through stimulation of the right DLPFC with rTMS might thus restore downstream striatal dysfunction in patients with AUD, reduce craving, and subsequently the risk for alcohol use. In contrast with this top-down strengthening of cognitive control, a recent study using deep TMS targeting striatal and lower frontal areas suggests that reduced bottom-up pass-through of impulses due to reduced connectivity between the striatum and anterior cingulate cortex might also reduce alcohol use [69]. Future studies should further explore working mechanisms of rTMS in AUD patients, for instance, combining rTMS treatment with neuro-imaging or electroencephalography measures. 

This study has to be evaluated in light of some strengths and limitations. Major strengths of the current study include the robust rTMS stimulation protocol of ten sessions and 30.000 pulses in total, and a long follow-up period of a year, with no drop-out in either treatment group. This provides insight into the potential time-dependent effect of rTMS. Though the sample size in the current study (*n* = 30) is substantially larger than the mean sample size for clinical studies in this field (*n* = 22.2) [70], it is still relatively modest. Small samples increase the risk of type I/II errors and might inflate the magnitude of effect (winner’s curse) [71]. Though MANOVA might overestimate due to correlation between outcome measures, this is unlikely to fully explain the effects observed here, given the low to moderate correlations between outcome measures. 

Medication use at baseline did not differ between groups. However, the current study did not account for benzodiazepine and antipsychotic use in follow-up. Though benzodiazepines were generally fully tapered off during detoxification, any persisting effect of benzodiazepine use on rTMS effectiveness cannot be ruled out. Similarly, antipsychotics were frequently prescribed (32%). Since it is known that benzodiazepines and antipsychotics may attenuate rTMS effects, the current findings might underestimate the effectiveness of rTMS in AUD [72]. However, augmentation of rTMS effects through simultaneous use of pharmacotherapy has also been reported [73].

The rTMS methods to determine MT and position the coil applied here are in line with FDA guidance for rTMS procedures [74]. However, studies suggest that other approaches might be more precise in providing the optimal personalized rTMS dose at the optimal personalized brain area by applying EMG measures and fMRI guidance, respectively [75,76]. Future studies might explore whether this is also the case in AUD treatment. 

Finally, the sham condition applied here (two wing 90-degree tilting of the coil) might be discernable by patients from real rTMS because it does not induce sensations on the scalp. This might have diminished the placebo effect in the control group. However, because major effects were found at three months follow up, confounding is unlikely because of this potentially reduced placebo effect. Future studies should ask patients about the expected group membership (real or sham rTMS) and/or apply other forms of sham rTMS; however, sham TMS approaches are inherently insufficient [77,78].

## 5. Conclusions

This small-scale randomized trial shows the potential efficacy of high-frequency rTMS applied over the right DLPFC on top of TAU in recently detoxified AUD patients on both alcohol craving and consumption. Given the observed effects were most prominent at three months follow-up, future studies should explore whether prolonged rTMS treatment or the use of booster sessions can induce more prolonged effects. Furthermore, future studies should explore potential working mechanisms and replicate these findings in substantially large clinical samples.

## Figures and Tables

**Figure 1 jcm-11-00951-f001:**
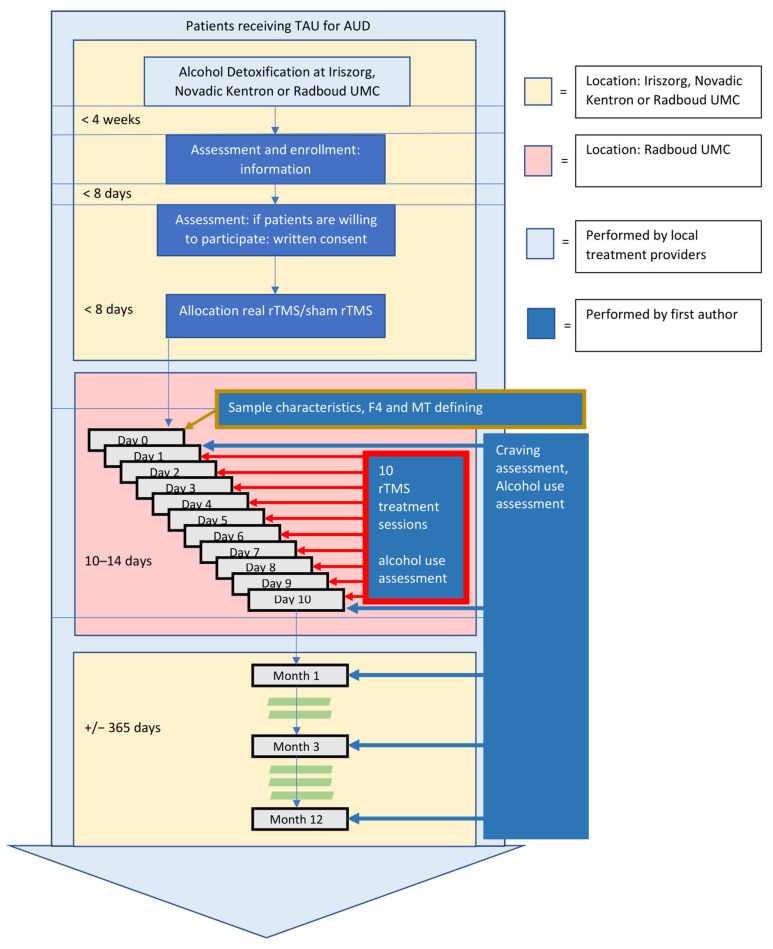
Schematic overview of the study procedure. TAU = Treatment As Usual; AUD = Alcohol Use Disorder; rTMS = repetitive Transcranial Magnetic Stimulation.

**Figure 2 jcm-11-00951-f002:**
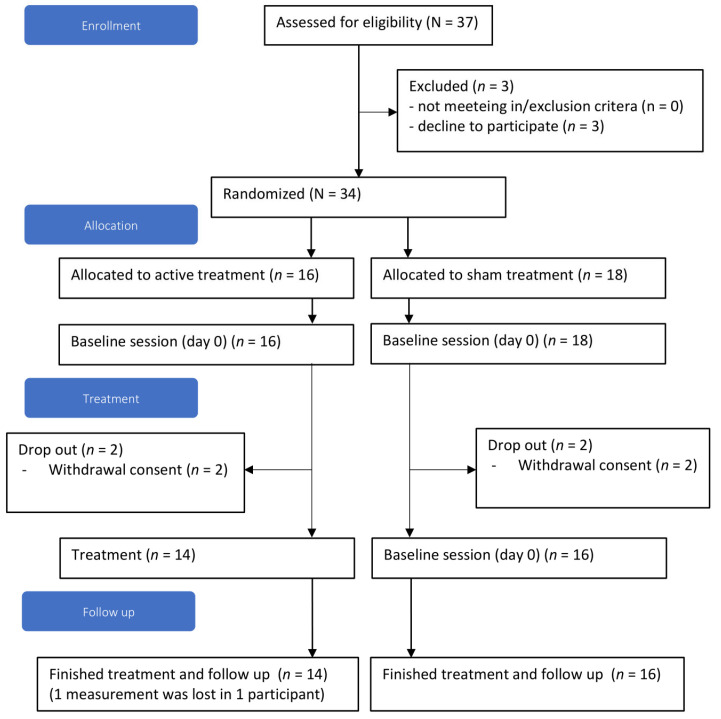
Consort flowchart.

**Figure 3 jcm-11-00951-f003:**
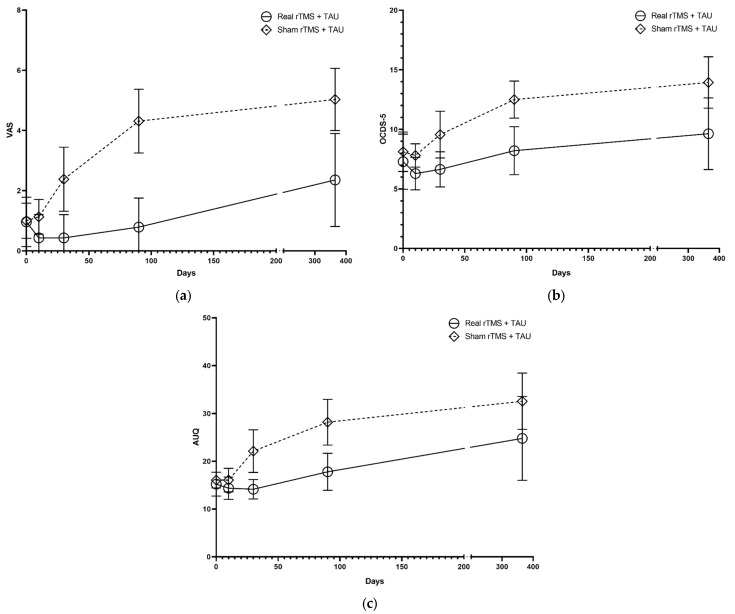
Effect means with 95% CI of rTMS on outcome measurements over time for (**a**) Visual analog scale (VAS), (**b**) Obsessive-compulsive drinking scale (OCDS-5), and (**c**) Alcohol urge questionnaire (AUQ).

**Table 1 jcm-11-00951-t001:** Baseline Sample Characteristics.

Variables	rTMS + TAU (*n* = 16)	TAU (*n* = 18)	Total Sample (*n* = 34)
Demographics			
Age, M (SD)	49.3 (7.9)	45.8 (9.7)	47.4 (8.9)
Gender, %male	100	89	94
IQ score, M (SD)	95.8 (14.5)	97.6 (14.7)	96.8 (14.4)
AUD, M (SD)			
Age first ever alcohol use	12.3 (3.5)	13.3 (3.4)	12.8 (3.4)
Years of problematic use	16.4 (6.5)	14.3 (7.4)	15.7 (7.0)
Consumption alcohol (gr/day)	132 (54)	122 (58)	127 (56)
Number previous treatments	3.8 (0.3)	4.6 (1.0)	4.2 (1.3)
Other substance use disorders			
Tobacco, %	81	94	88
Cannabis, %	0	6	3
Stimulants, %	0	6	3
Benzodiazepine, %	0	6	3
Psychiatric comorbidity			
PTSD, %	0	44	26
Depression, %	13	17	15
OCD, %	0	22	12
Panic disorder, %	0	0	0
Measurements baseline			
VAS, M (SD)	0.9 (1.4)	0.8 (1.2	0.9 (1.2)
OCDS, M (SD)	7.3 (4.0)	8.1 (3.1)	7.7 (3.5)
AUQ, M (SD)	15.9 (5.3)	14.3 (5.9)	15.1 (5.7)
Abstinent, %	100	94	97
Alcohol use (gr/day), M (SD)	0 (0)	0.13 (0.13)	0.07 (0.03)
Heavy drinking, M (SD)	0 (0)	0 (0)	0 (0)
Use of medication			
Anticraving, %	6	6	6
Antidepressants, %	19	44	32
Antipsychotics, %	19	44	32
Benzodiazepines, %	50	72	62

M = Mean; SD = Standard Deviation; AUD = Alcohol Use Disorder; PTSD = Post-traumatic stress disorder; OCD = Obsessive compulsive disorder; VAS = Visual Analog Scale; OCDS = Obsessive Compulsive Drinking Scale; AUQ = Alcohol Urge Questionnaire

## Data Availability

The data presented in this study are openly available in FigShare at doi 10.6084/m9.figshare.19146284.

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
