# Peer review of "rTMS Reduces Craving and Alcohol Use in Patients with Alcohol Use Disorder: Results of a Randomized, Sham-Controlled Clinical Trial"

_jcm, 2022, doi:10.3390/jcm11040951_

Round 1

Reviewer 1 Report

This is a small, but nicely designed and reported study of “high-dose” rTMS to the right dlPFC for alcohol use disorder. Alhough they will need to be replicated in larger samples, the results are promising both for craving reduction and reduction of use up to 3 months after completion of treatment. Please address the issues raised below.

--

“AUD is associated with a decreased life expectancy of 24-28 years compared 37 to the general population [4].”  - this is misleading – these numbers refer to a subpopulation of the most severe AUD, with drinking in excess of 100 g/day (males) or 60g/day (females).  

“Specifically, we tested the hypotheses that compared to sham rTMS, rTMS over the right dlPFC, added to treatment as usual (TAU), would lead to reduced 1) alcohol craving (primary outcome), and 2) alcohol use (secondary outcome).” – this is a bit problematic, as craving is not accepted (neither by the EMA nor the FDA) as a “meaningful benefit”; abstinence, or suffiently large decreases in use are. This needs to be discussed, although is less of a problem given that both outcomes seemed to indicate a response.

For eligibility criteria, please specify how the diagnosis of DSM-5 AUD was established. Also, was really any severity of AUD included? Not just moderate – severe, i.e. corresponding to ICD / DSM-IV Alcohol Dependence?

It is stated that TAU could include medications. Please include in the table on baseline characteristics frequency data on on main classes of CNS active medications including anticraving meds, to ensure that there is no inadvertant imbalance between the randomization groups that could confound the results.

The methods indicate that the study was blinded for the participants, and how this was achieved. However, the way this was done means that the sensation is very different for sham compared to placeby (there is no twitching of the scalp muscles). It is therefore critical to know to what extent participants were, or were not able to see through the blind.

This may not be a big issue if it is really true that only one datapoint for one participant was missing; but LOFC is not considered a valid method of imputating missing values in clinical trials. Please see current regulatory guidelines. The reason is that data are almost never truly mssing at random. The standard approach to evaluating repeated measures designs where some data are missing, with the least (but non-zero) potential for introducing a bias is a MMRM-approach with subjct as random effect, and no imputation. Data in the trial will have to be re-aevaluated accordingly. Only then will it be possible to determine what the result is.

The MANOVA approach is potentially problematic, because the 3 measures of craving are almost certain to be highly correlated. In an exploratory study like this one, it migth be OK to evaluate which craving score that has the highest sensitivity (which the authors separately report); create some kind of composite score; or, alternatively, run a PCA, and use the factor scores as a dependent measure. The MANOVA may overestimate effects due to correlation of the measures. The same goes for the alcohol measures. Also, time to lapse or relapse are not suitable for this type of analysis, they are classical survival analyses that should be evaluated using Cox-regression.

“In case of significant results, post-hoc contrast analyses were performed to identify which craving outcome measures at which specific timepoints contrib to the significant findings.” – unclear what type of post-hoc test, whether one that protects experiment-wise error rats for multiplicity of testing

In their discussion, the authors may want to include the Harel et al paper on rTMS targeting the mPFC / ACC in AUD that has meanwhile come out in Biological Psychiatry

Author Response

Dear Ms Gao

We appreciate the highly instructive comments of the reviewer and have adjusted the manuscript accordingly. In particular, suggested by the reviewer, the manuscript includes additional information about clinical characteristics of the patients and details of the rTMS sessions, as well as results from correlational analyses.

We think that the article has improved substantially by the comments and suggestions, each of which is specifically addressed below. We have highlighted the changes in the manuscript.

We would like to thank the reviewers for their remarks and would like to address them below.

Reviewer 2 Report

Even though the study had an interesting aim, the problem is with the methodology. Firstly, the authors used a TMS device without EMG and without neurophysiological testing. No excitability measure was done to be associated with qualitative outcomes (self-report scales). The Resting motor threshold  (RMT) was determined by visual detection of muscle contraction, therefore the 110% intensity used for rTMS treatment might have been higher than by recommended procedures. An extensive psychological assessment would be highly necessary to be done for these patients for baseline and for rTMS sessions that might be associated with outcome measures used in the present study. The author did not report if they performed a psychological evaluation of patients.

Methods

-The resting motor threshold is calculated by observing visually hand muscle contractions which is not the proper way how MT should be done. The author obviously did not use EMG. 110% intensity on MT (determined by visual muscle contraction) is somehow higher intensity versus classical standard way how MT is done.

-Please explain more clearly the sentence: “A total of 30,000 pulses are amongst the highest number used in studies in this field, while being well below threshold of increasing risks on side-effects [23, 24].

-3000 pulses per day - is this protocol formally allowed regarding the patient safety guidelines for rTMS use in addition

-Please write the time needed per one rTMS session

-does the patient get a hearing sponge during rTMS sessions?

-Does the patients have a psychological evaluation before enrolling into rTMS? This testing and picking special instruments to have at the baseline and during follow-up treatments would be beneficial.

Discussion

-What about other rTMS stimulating protocols other than 10 Hz like theta bursts? Discussion on this matter should be performed in the alcohol addition field.

Neuron, Vol. 45, 201–206, January 20, 2005, Copyright ©2005 by Elsevier Inc. DOI 10.1016/j.neuron.2004.12.033

Theta Burst Stimulation Report

of the Human Motor Cortex

-row 373-375 sentence please explain more clearly

Author Response

Dear Ms. Gao

We appreciate the highly instructive comments of the reviewer and have adjusted the manuscript accordingly. In particular, suggested by the reviewer, the manuscript includes additional information about details of the rTMS sessions, safety concerns, as well as added discussion.

We think that the article has improved substantially by the comments and suggestions, each of which is specifically addressed below. We have highlighted the changes in the manuscript.

We would like to thank the reviewers for their remarks and would like to address them below.

Round 2

Reviewer 1 Report

Although the paper is improved, several issues remain:

-As per regulatory guidance from both the EMA and the FDA, LOCF is not an appropriate method for imputing missing data. This is because it assumes that data are missing at random, while that is almost never the case (and in particular not in addiction studies, where the most likely reason for a missed visit is relapse). Although this is not a trivial issues, the most commonly recommended approach is a mixed model analysis with subject as a random effect, using all data available, and no imputation

-The authors continue to argue that craving and as well as reduction of amount alcohol consumed are valid outcome measures. Once again, regulatory guidance from both t he EMA and the FDA says otherwise. The EMA is moving a bit faster on this, and has indicated it might consider a drop by 2 WHO drinking level classes as "meaningful clinical benefit", but that is it. Therefore, the most the current paper can be said to do is to provide biomarker evidence suggesting that a future, adequately powered and designed study, might find effects on accepted clinical outcome measures. Any claims beyond that are unwarranted in view of the regulatory realities, and cannot be continued.

Author Response

Dear Ms. Gao,

Thank you for your interest in our study “rTMS reduces craving and alcohol use in patients with alcohol use disorder: Results of a randomized, sham-controlled clinical trial”.

Again, we appreciate the instructive comments of the reviewers and have adjusted the manuscript accordingly.

We would like to thank the reviewer for the remarks and will address them in the attached word document.

Reviewer 2 Report

-row 301, please check/correct the acronym SUD? I believe the authors intended to write AUD.

-the reader might find too many acronyms in the text, it would be suggested to write the abbreviation paragraph before the introduction section

-it is also suggested that authors include in Discussion the limitation related to the methodology they used (intensity issue regarding the RMT determination) and the NBS (navigated brain stimulation) that was not used in this study. It might be interesting to conduct the same study but compare the effect of NBS - rTMS. It is not sufficient just to write on other studies including the reference as the only argument why the methodology in this study was optimal. It is reasonable to believe that inclusion of individual MRI, with NBS-rTMS with proper determination of RMT (not observing visually the muscle contraction in the hand muscles) would have more acceptable results.

-One would also think regarding the limitation to discuss the psychological tests determining the personality profile and to see whether the effects of rTMS (and duration) have any association with a personality profile.

-Could the author include in the Introduction or discussion if tDCS studies were used in the field of alcohol craving and possible effect differences between tDCS and rTMS

-Could the author include in the Introduction or discussion if tDCS studies were used in the field of alcohol craving and possible effect differences between tDCS and rTMS

Author Response

(The authors gave the same response as above.)
